# Propulsion Mechanism of Flexible Microbead Swimmers in the Low Reynolds Number Regime

**DOI:** 10.3390/mi11121107

**Published:** 2020-12-15

**Authors:** Yan-Hom Li, Shao-Chun Chen

**Affiliations:** Department of Mechanical and Aerospace Engineering, Chung-Cheng Institute of Technology, National Defense University, Taoyuan 335, Taiwan; weekend123@gmail.com

**Keywords:** microswimmer, superparamagnetic microbeads, flagellum, low Reynolds number

## Abstract

A propulsion mechanism for a flexible microswimmer constructed from superparamagnetic microbeads with different diameters and subjected to an oscillating field was studied experimentally and theoretically herein. Various types of artificial swimmers with different bending patterns were fabricated to determine the flexibility and an effective waveform for a planar beating flagellum. Waveform evolutions for various swimmer configurations were studied to determine the flexible mechanism of the swimmers. A one-armed microswimmer can propel itself only if the friction of its wavelike body is anisotropic. A swimmer with a larger head and a stronger magnetic dipole moment with a flexible tail allows the bending wave to propagate from the head toward the tail to generate forward thrust. The oscillating head and tail do not simultaneously generate positive thrust all the time within a period of oscillation. To increase the propulsion for a bending swimmer, this study proposes a novel configuration for a microbead swimmer that ensures better swimming efficiency. The ratio of the oscillation amplitude of the head to the length of the swimmer (from 0.26 to 0.28) produces a faster swimmer. On the other hand, the swimmer is propelled more effectively if the ratio of the oscillation amplitude of the tail to the length of the swimmer is from 0.29 to 0.33. This study determined the optimal configuration for a flexible microbead swimmer that generates the greatest propulsion in a low Reynolds number environment.

## 1. Introduction

At a low Reynolds number, any reciprocal motion that involves symmetric backward and forward movement does not generate net thrust because it is a time-reversible motion [1], which is commonly referred to as the scallop theorem [2]. To allow locomotion for a microswimmer in a low Reynolds number environment, various magnetic actuation methods have been used to generate a higher propulsive efficiency to allow movement in a viscous fluid. Three main propulsion mechanisms that are inspired by microorganisms have been developed to transport microscopic particles in a viscous fluid using uniform magnetic fields [3,4]. 

The first group is the surface walkers, which rely on nearby surfaces to break the spatial symmetry and to provide one additional degree of freedom, and they defy the scallop theorem to generate propulsion [5,6,7,8,9,10]. The second group of artificial swimmers is propelled by helical tails that are controlled using a rotating magnetic field [11,12,13,14,15]. The third group is the one-armed swimmers that are referred to as flexible propellers and generate propulsion using a flexible flagellum fabricated using a DNA linkage [16] or nanowire [17,18], actuated by an oscillating magnetic field. In recent cases, particle-based magnetic microswimmers were fabricated and manipulated under a rotating magnetic field [19,20,21,22,23,24,25,26]. Other proposed methods to develop novel magnetic actuation sperm-like microswimmers have also been demonstrated [27,28,29]. MagnetoSperm consists of an ellipsoid head and a trapezoidal tail to mimic the shape of a sperm cell, which was demonstrated by Khalil et al. [27], who also developed and controlled a soft robotic sperm that undergoes controllable switching between planar and helical flagellar swimming [28]. Recently, Magdanz et al. constructed biohybrid magnetic microrobots by electrostatic self-assembly of nonmotile sperm cells and magnetic nanoparticles [29].

An undulating microswimmer that is fabricated from superparamagnetic microbeads can be regarded as the simplest mechanism to duplicate because it has a one-dimensional structure. A recent experimental study demonstrated steering for a directed assembly swimmer that is fabricated directly from multiple microbeads without any chemical link and is subjected to an oscillating field [30]. A physical mechanism for the flexibility of a particle-based flagellum [31] and swimmer [32] has been described. The self-assembling ability of such dipolar colloids allows their fabrication and contactless manipulation, but it is difficult to control a stable flexible microbead swimmer without any chemical link and to achieve optimal propulsive efficiency at the microscale. In addition, the waveform pattern that is caused by the undulatory motion of a one-armed swimmer has a significant effect on swimming efficiency [33,34]. Optimal flexible configuration to propagate a bending wave is crucial for the propulsion of a bead-based swimmer and its further applications.

This study conducted a series of experiments to model undulating microbead swimmers as a continuous elastic flagellum and determined the propulsion mechanism for a one-armed swimmer with various waveforms. Additionally, the waveform that generates the most effective propulsion generation and the dimensionless factors for the optimal configuration for a swimmer with high propulsive efficiency are also proposed.

## 2. Materials and Methods

Figure 1 shows the experimental configuration for this study. A horizontally homogeneous static field, H_x_, was generated using a pair of coils that are powered by a DC power source (Guan Chun, New Taipei City, Taiwan). The magnetic beads that are magnetized by the static field *H*_x_ tended to aggregate and randomly form various types of chain-like swimmers. A perpendicularly dynamic sinusoidal field, *H*_y_, with a maximum amplitude, *H*_p_, and an adjustable frequency, *f*, such that *H*_y_ = *H*_p_ sin(2π*ft*), was generated using a pair of coils that are connected to an AC power supply (GWInstek APS-1102, GWInstek, New Taipei City, Taiwan) to constitute an overall oscillating field.

To generate locomotion for simple geometry microswimmers in an environment with a low Reynolds number, swimmers were constructed using an artificial smart fluid that contained two types of self-assembling superparamagnetic particles. These particles consisted of iron oxide magnetite (Fe_3_O_4_), which was embedded in polystyrene microspheres that were suspended in distilled water (Thermo Fisher Scientific Inc., Waltham, MA, USA: Dynabeads M-450 Epoxy and Dynabeads M-270 Epoxy). The mean diameters of the microparticles were *d* = 4.5 and 2.8 μm, and the respective magnetic susceptibilities were χ = 1.6 and 1.0. The microbeads had a saturation magnetization of M = 28,000–32,000 A/m and did not exhibit magnetic hysteresis or remanence, so these magnetic beads could be linearly magnetized or de-magnetized by applying or removing an external field. When the external applied field was removed, the microbeads dispersed again. The movement of the beads and the swimmers was recorded using an optical microscope that was connected to a high-speed camera with a maximum rate of capture of 200 frames per second.

## 3. Results and Discussion

### 3.1. Non-Reciprocal Motion for a Microbead Swimmer

To generate a net thrust for a one-armed microswimmer, symmetrical oscillation must be prevented. Figure 2 shows four types of time-irreversible microswimmers that were subjected to an identical field strength of *H*_p_ = 2000 A/m and *H*_d_ = 1900 A/m that oscillates at a frequency of *f* = 10 Hz. All swimmers consisted of different numbers of magnetic microparticles of different sizes. Some previous investigations on the effect of the frequency have shown that a microbead chain or swimmer can be manipulated with a stable structure or efficient movement at a frequency of 7–10 Hz [35,36]. Therefore, all the swimmers were subjected to an oscillating frequency of *f* = 10 Hz in this study.

Figure 2a shows a flexible pattern for a swimmer composed of one large and three small microbeads (denoted as L1S3). Figure 2b–d show the sequential images of the L2S2, S1L1S3, and L3S1 swimmers, respectively, oscillating within an arbitrary period (denoted as P). Within a period of oscillation, the L1S3, L2S2, and S1L1S3 swimmers evolved from a rigid state to distortion and then returned to the rigid state. Meanwhile, the L3S1 swimmer exhibited an insignificant deformation because it featured a nearly symmetrical geometry that showed a slight distortion at its short tail. It is noted that magnetic swimmers that are made of microbeads without a chemical link, such as DNA [16] or streptavidin–biotin binding [19,21] do not retain a stable S-shaped structure under the influence of a higher oscillation frequency. These easily fabricated flexible swimmers have a short, curved shape with limited length, so they generate only limited flexible structure in a strong field strength that oscillates at a moderately high frequency. In spite of having less flexibility than those with a chemical link, their asymmetrical structure generates the non-reciprocal asymmetric motion that is necessary to generate propulsion for a microbead swimmer. 

The perpendicular moving trajectories of each particle of each swimmer in Figure 2 were measured and are depicted in Figure 3. The points on each curve represent the center of mass of the microbeads that form the swimmer. The crossover area for all curves indicates the center of mass of the swimmer. The number of the curve is marked on the side of the head or the swimming direction of the swimmer. Curve No. 1 in Figure 3a represents the nearly rigid L1S3 swimmer, as it starts from the top in a near stagnation position and is about to oscillate in a clockwise direction. The swimmer was subject to a magnetic force and an opposing hydrodynamic drag during oscillation. There was a stronger induced magnetic dipole moment at the head because it had a greater magnetic susceptibility of χ = 1.6, so the L1S3 swimmer’s head was driven to move faster than the tail. Each particle rotated around the center of mass of the swimmer during oscillation. The distance from the head to the center of mass of the swimmer was shorter than the distance from the tail to the center of mass of the swimmer. Thus, to maintain the shape of a rigid beam, the particle closest to the tail must move faster than the particle closest to the center of mass of the swimmer. However, the hydrodynamic drag delayed the movement of particles so the particle closest to the tail lagged behind the head. The last particle of the tail lagged more significantly than the others. As a result, the swimmer flexed as a wave, as shown by curve No. 2. 

Curve No. 3 shows that the head of the swimmer almost achieved the lowest position and that the tail particle still lagged behind the head. Curve No. 4 represents the bottom stagnation position, where the tail caught up with the head and the swimmer exhibited an almost rigid shape again. For a reverse (counterclockwise) oscillation, curves No. 5–No. 7 show the same scenario. Figure 3b–d show the waveform evolutions for the L2S2, S1L1S3, and L3S1 swimmers, respectively. The L2S2 and S1L1S3 swimmers demonstrated a waveform evolution similar to that for the L1S3 swimmer. However, an insignificant curve shape emerged for the L3S1 swimmer because of its slightly asymmetric geometry. The results in Figure 3 also show that there was a difference in the oscillation amplitude of the head and tail for the various types of swimmers. These results were caused by their different lengths and flexible structures and would lead to various propulsive velocities, which is demonstrated in the following sections.

### 3.2. Propulsion Mechanism for a Swimmer

In a low Reynolds number environment, inertial force is negligible, so the propulsion is only generated by an undulating motion if the propelling body demonstrates frictional anisotropy with the embedding fluid. Figure 4 shows a diagram of the net thrust generation mechanism for a microswimmer in a low Reynolds number environment. In Figure 4a, the tail of the microswimmer exerts a force on the fluid (black arrow) when it pushes the fluid, so the fluid exerts a reaction force on the microswimmer (light blue arrow). This reaction force can be decomposed into normal (N) and tangential (T) components (green arrow). The net thrust is the difference between the projection of T and N (red) along the axis of thrust, as shown in Figure 4b, i.e., Nsin*θ* − Tcos*θ*. A greater frictional force is required to push a swimmer perpendicular to its elongated body than to push parallel to it. The same scenario occurs at the head of the microswimmer. As a consequence, a microswimmer generates propulsion only if the friction of its wavelike body or head is anisotropic. Theoretically speaking, the greater the oscillation or the angle *θ* of the undulating swimmer, the greater the net thrust.

The undulating motion that is associated with anisotropic friction is necessary to generate propulsion for a swimmer, so the mechanism to induce an effective undulating waveform is important to producing more effective propulsion. The shape of a swimmer is most significantly affected by the magnetic force and induced drag, so there is a complex acceleration and deceleration process that significantly influences the deformation of a swimmer. The motion of an oscillating swimmer is theoretically classified into four stages within a period of oscillation, as shown in Figure 4c. At Stage 1, the swimmer begins to oscillate and accelerates from the top stagnation point. During this acceleration, the swimmer encounters an opposing drag force, so it deforms, as shown in Figure 2. As the swimmer passes through the horizontal axis of thrust, it decelerates toward the bottom stagnation point in Stage 2. The swimmer then oscillates in the opposite direction and experiences acceleration during Stage 3 and deceleration during Stage 4. This constitutes a full period of oscillation. 

### 3.3. The Waveform That Generates Effective Propulsion for a Swimmer

The head and tail of the microbead oscillate with the external field because they exhibit superparamagnetic properties. Larger particles at the head with a higher susceptibility of χ = 1.6 oscillate faster than the tail, so a bending wave travels from head to tail, which results in the flexibility and frictional anisotropy of a swimmer. However, the oscillating head and tail do not simultaneously generate positive thrust for the entire period of oscillation.

Figure 5 shows the direction of propulsion for the head and tail of a swimmer within a half period of oscillation. At the initial position in Figure 5a, the swimmer accelerates in a clockwise direction and the head and tail generate negative and positive thrust, respectively, for a swimming direction to the right. Figure 5b shows that the swimmer distorts because the head oscillates faster than the tail. However, this slight distortion has no effect on the direction of propulsion for the head and tail, so the propulsion is not efficient. When the head passes the horizontal axis of thrust, the swimmer is subject to significant deformation, so there is a change in the direction in which the head is propelled. At the stage as shown in Figure 5c, the head and tail simultaneously generate positive thrust, which demonstrates the waveform that most effectively generates propulsion. The head thereafter generates a positive thrust, as shown in Figure 5d until the swimmer reaches the bottom stagnation position. However, the tail begins to propel opposite to the swimming direction when it passes the axis of thrust, so propulsion efficiency is predicted to decline slightly at this stage. There is a similar scenario when a swimmer begins to oscillate in the opposite direction.

A study of the direction of propulsion for the head and tail within a period of oscillation shows that both the head and tail generate positive and negative propulsion in this specific period. To enhance the overall propulsive efficiency of a microbead swimmer, the combination of the microbeads and configuration of the swimmer need to be optimized. There are two possible methods to achieve this goal. First, increase the dipole magnetic moment at the head and increase the length of the flexible tail, so that the swimmer is able to distort to a greater degree and create a more effective waveform. Second, to shorten the head to reduce negative thrust during the head’s acceleration stage, as shown in Figure 5a,b.

Figure 6 shows the direction of thrust for the head and tail for various configurations of swimmers within a period of oscillation. Image 1 shows all the swimmers at the top stagnation position. The swimmers do not generate a net thrust because their bodies are rigid. In images 2 and 3, the swimmers experience acceleration and deceleration, which produces flexibility and anisotropic friction, and they generate a net positive thrust that produces locomotion. In images 4–7, the swimmers are subject to the second half of oscillation, which also generates net thrust via flexibility and anisotropic friction. The images in Figure 7 show that each swimmer has its own effective waveform at a specific stage. The effective waveform causes the head and the tail to generate positive thrust simultaneously, which only occurs when the oscillating head passes the axis of thrust, as shown in Figure 5c.

In Figure 6, the L1S3, L2S2, and S1L1S3 swimmers generated the effective waveforms that are shown in images 2, 3, and 6. In contrast, the effective waveform of the L3S1 swimmer shown in image 6 exhibited insignificant flexibility. As a result, the different configurations led to variations in the net thrust of the swimmers. A comparison of the swimming effectiveness for the different types of swimmer is demonstrated in the next section.

### 3.4. Theoretical Equation for the Net Thrust of the Microbead Swimmer

To theoretically study the propulsive force of a swimmer, a model was modified to calculate the forces acting on each microbead of the swimmer based on the resistive force theory of sperm motion [37]. The forces that act on one of the individual beads should reflect the forces imposing on the fluid. Figure 7 illustrates the forces imposed on the *i*th particle (denoted as *P_i_*) when the swimmer moves along the axis of propulsion (x-axis). The orientation (*θ_i_*) of the *i*th particle to the axis of propulsion (x-axis) depends on the form of the wave and the particle’s position on the wave. As shown in Figure 7a, the velocity of the *i*th particle along the y-axis is *V_i,y_* and its surface is inclined to the x-axis by an angle of *θ_i_.* The transverse velocity of *V_i_,_v_* consists of two components: A tangential velocity, *V_i,y_*sin*θ_i_,* and a normal velocity, *V_i,y_*cos*θ_i_*. The fluid imposes tangential (*T_i,y_*) and normal (*N_i,y_*) reactions on the surface of the *i*th particle during the transverse motion because of the resistance of the fluid. The forces *N_i_,y* and *T_i,y_* have the components *N_i_,y*sin*θ_i_* and *T_i_,y*cos*θ_i_*, respectively, acting along the axis of propulsion. Thus, the resultant propulsive thrust *F_i,y_* along the axis of propulsion is given as:(1)$$F_{i,y}=N_{i,y}\sin\theta_{i}-T_{i,y}\cos\theta_{i},$$

Since the particle is extremely small and the moving speed is very low, the reaction forces on the particle derived from the fluid can be regarded as directly proportional to the moving velocity and the viscosity of the fluid:(2)$$N_{i,y}=C_{N}V_{i,y}\sin\theta_{i}d_{i},$$
(3)$$T_{i,y}=C_{T}V_{i,y}\cos\theta_{i}d_{i},$$
where *C_N_* and *C_T_* are the coefficients of resistance to the surface of the *i*th particle for a fluid of known viscosity, and *d_i_* is the diameter of the *i*th particle. As a result, the resultant forward thrust *F_i,y_* along the axis of propulsion is given as:(4)$$F_{i,y}=(C_{N}-C_{T})V_{i,y}\sin\theta_{i}\cos\theta_{i}d_{i},$$

Thus, the transverse movement of the particle generates a forward thrust along the axis of propulsion. Moreover, each particle in a microbead swimmer also moves along the axis of propulsion at a speed of *V_x_,* which depends on the speed of the whole swimmer moving through the fluid. Figure 8b shows that the resultant drag is composed of *N_i,x_*sin*θ_i_* and *T_i,x_*con*θ_i_*, which are directly proportional to the velocity of the displacement and the viscosity of the fluid. As a result, the resultant drag *F_i_,_x_* along the axis of propulsion is given as:
(5)$$F_{i,x}=(C_{N}\sin^{2}\theta_{i}+C_{T}\cos^{2}\theta_{i})V_{i,x}d_{i},$$

The net propulsive thrust caused by the simultaneous transverse and forward movement is obtained by combining Equations (4) and (5). The total force can also be expressed by the normal and tangential forces acting to the surface of the *i*th particle due to its transverse and forward motion, which are respectively given as:(6)$$N_{i}=C_{N}(V_{i,y}\cos\theta_{i}-V_{i,x}\sin\theta_{i})d_{i},$$
(7)$$T_{i}=C_{T}(V_{i,y}\sin\theta_{i}+V_{i,x}\cos\theta_{i})d_{i},$$

The propulsive components of *N_i_* and *T_i_* along the axis of propulsion are expressed as *N_i_*sin*θ_i_* and *T_i_*cos*θ_i_,* respectively, and the net forward thrust of the *i*th particle (*F_i_*) is:(8)$$\begin{array}{l} F_{i}=N_{i}\sin\theta_{i}-T_{i}\cos\theta_{i}=F_{i,y}-F_{i,x}\\ =[(C_{N}-C_{T})V_{i,y}\sin\theta_{i}\cos\theta_{i}-V_{i,}{}_{x}(C_{N}\sin^{2}\theta_{i}+C_{T}\cos^{2}\theta_{i})]d_{i}, \end{array}$$

The overall forward thrust *F_t_* for a swimmer consisting of *N* particles at specific waveform is then given as:(9)$$F_{t}=\sum_{i=1}^{N}[(C_{N}-C_{T})V_{i,y}\sin\theta_{i}\cos\theta_{i}-V_{x}(C_{N}\sin^{2}\theta_{i}+C_{T}\cos^{2}\theta_{i})]d_{i},$$

### 3.5. Comparison of the Propulsive Speed of Various Types of Microswimmers

The locomotion of microbead swimmers with various configurations is shown in Figure 8. All swimmers were manipulated in a field for which *H*_p_ = 2000 A/m and *H*_d_ = 1900 A/m, and there was a constant frequency of *f* = 10 Hz. The swimmers swam toward the direction of their center of mass, which was located closer to the larger particles, so they moved to the right in Figure 8a,b and to the left in Figure 8c,d. The propulsive speed was measured by recording the moving trajectory of the swimmer. A comparison of the swimming effectiveness for the four swimmers is shown in Figure 9. It can be seen that the L2S2 swimmer moved farther than the rest of the other swimmers in 10 s. The maximum respective average velocities for the L1S3, L2S2, S1L1S3, and L3S1 swimmers were 1.38, 2.28, 1.55, and 0.9672 μm/s. These values of speed reveal that the swimmers can swim in an environment with a low Reynolds number of 1.77 × 10^–5^–3.74 × 10^–5^.

The reason for the significant difference in the velocity can be understood by Figure 6 and Figure 10, which show effective waveform and the configuration, respectively, of the four swimmers. In Figure 10a, the L3S1 swimmer is the longest and has the largest head. The longer body is intuitively in favor of propulsion generation. However, the nearly rigid body and the shorter deformed tail failed to form a significantly flexible structure, so there was a lack of anisotropic friction. Moreover, the larger head resulted in more ineffective thrust for the locomotion. As a result, L3S1 was the slowest among the swimmers. The other swimmers had a more flexible structure, because of the shortest length of the L1S3 swimmer, it was slower than the L2S2 and S1L1S3 swimmers. On the other hand, the L2S2 swimmer is slightly shorter than the S1L1S3 swimmer, but its head is composed of two larger beads, which are driven much faster by the external field and have a more significant flexible structure associated with a more effective waveform. As a result, the net projection of the normal and tangential force along the axis of thrust for the L2S2 swimmer was greater, which increased its propulsive efficiency. Based on the above experimental results, the L2S2 swimmer has the best configuration for propulsion generation, which requires a faster moving head and a flexible tail to allow the bending wave to travel in the opposite direction to its locomotion.

To verify this hypothesis, a microbead swimmer with a longer flexible tail was designed to compete with the L2S2 swimmer. Figure 11 shows the L2S2 and L2S3 swimmers in an identical field strength of *H*_p_ = 1500 A/m and *H*_d_ = 1900 A/m oscillating at a constant frequency of *f* = 10 Hz. *T* is the time when the swimmers began swimming at the highest speed and then cruised at a constant speed. Their moving trajectories are shown in Figure 12. The L2S3 swimmer traveled a farther distance than the L2S2 swimmer in 5 s. The average swimming velocity of the L2S3 swimmer was 2.24 μm/s, which is faster than the 1.9 μm/s for the L2S2 swimmer. These values of speed reveal that the swimmers can swim in an environment with a low Reynolds number of 3.12 × 10^–5^–4.38 × 10^–5^. The faster speed of the L2S3 swimmer is attributed to its longer flexible tail, which generates much more positive thrust within a period of oscillation. It is noted that the L2S2 swimmer was slower than the L2S2 case shown in Figure 8 because the dynamic field strength was lower at *H*_p_ = 1500 A/m, so there was a smaller dipole moment at the head and the amplitude and flexibility were decreased. A microbead swimmer with a larger head that is subject to a stronger induced dipole moment and that has a moderately long flexible tail generates the greatest thrust.

### 3.6. Crucial Factors for the Design of a Microbead Swimmer with Optimal Propulsive Efficiency

To define the best configuration for a microbead swimmer for propulsion generation, this study proposes the dimensionless factors *A*_h_/*L* and *A*_t_/*L* to determine the waveform characteristics for a flexible swimmer, and studies the effect of these factors on the propulsion efficiency of a swimmer. Figure 10b illustrates the definitions of *A*_h_/*L* and *A*_t_/*L*, which are the respective ratios of the amplitude of the head and the amplitude of the tail of the swimmer (denoted as *A*_h_ and *A*_t_, respectively) to the overall length of the swimmer (denoted as L). It has been shown that a larger head with a stronger dipole moment induces fast oscillation at the head and allows a wave to propagate from the head to the tail. A longer swimmer requires a larger (or longer) head to create a bending wave that gives greater propulsive efficiency. However, a larger (longer) head oscillates with a greater amplitude and produces much more negative thrust during the accelerating stage, as shown in Figure 5b. Effective locomotion requires a swimmer to have an oscillating head, but the amplitude is negatively correlated to swimming velocity.

A systematic experiment has been conducted to determine the configuration that most effectively generates propulsion for a microbead swimmer that is subject to different field strengths and frequencies. To provide a nondimensional propulsion speed for a swimmer, the original swimming velocity (*U*) was scaled off of the product of the body length (*L*) and the frequency (*f*), i.e., *Lf* [16]. The dimensionless propulsion velocity is expressed as *U/Lf*, which is the number of body lengths traveled per revolution of actuation. Figure 13 shows that the distribution of the dimensionless swimming velocity for various types initially increased with the ratio of A_h_/L and achieved its highest value between 0.015 and 0.02 for a value of *A*_h_/*L* = 0.26–0.28. This trend indicates that an increase in the oscillating amplitude initially enhances the propulsive efficiency of the swimmer. However, the dimensionless velocity begins to decrease when the value of *A*_h_/*L* is greater than 0.28, because a swimmer with a larger (longer) head generates greater negative thrust when the oscillating amplitude of the head is increased significantly. A similar trend was observed for the effect of the oscillating amplitude of the tail. Figure 13b shows that the value of the dimensionless velocity was greatest when the ratio of *A*_t_/*L* is 0.29–0.33. The dimensionless propulsion speed decreased thereafter as the value of *A*_t_/*L* increased. A higher value for *A*_t_/*L* than for *A*_h_/*L* produced the greatest propulsion, which implies that a one-dimensional magnetic microbead swimmer with a tail of higher beating amplitude dominates the effective locomotion in a low Reynolds number environment.

## 4. Conclusions

Various types of microswimmers were designed in this study using superparamagnetic microbeads of different sizes and properties. These microbead swimmers move at different velocities in an identical oscillating field. The mechanism for propulsion generation and the key factors affecting the swimmers’ propulsive efficiency were determined. A larger head with a stronger induced dipole moment allowed a bending wave to propagate from the head to the tail, which induced a flexible structure and anisotropic friction and generated a net positive thrust. 

Two crucial factors for the design of a microbead swimmer with higher propulsive efficiency were proposed. An oscillating head amplitude to swimmer length ratio (*A*_h_/*L*) of 0.26 to 0.28 produced the greatest swimming velocity. In contrast, a swimmer was faster when the ratio of the oscillating amplitude of the tail to the length of the swimmer (*A*_t_/*L*) was 0.29 to 0.33. This study determined the optimal configuration for a flexible microbead swimmer that produces the greatest swimming effectiveness, which can potentially be applied in a microfluidic system.

## Figures and Tables

**Figure 1 micromachines-11-01107-f001:**
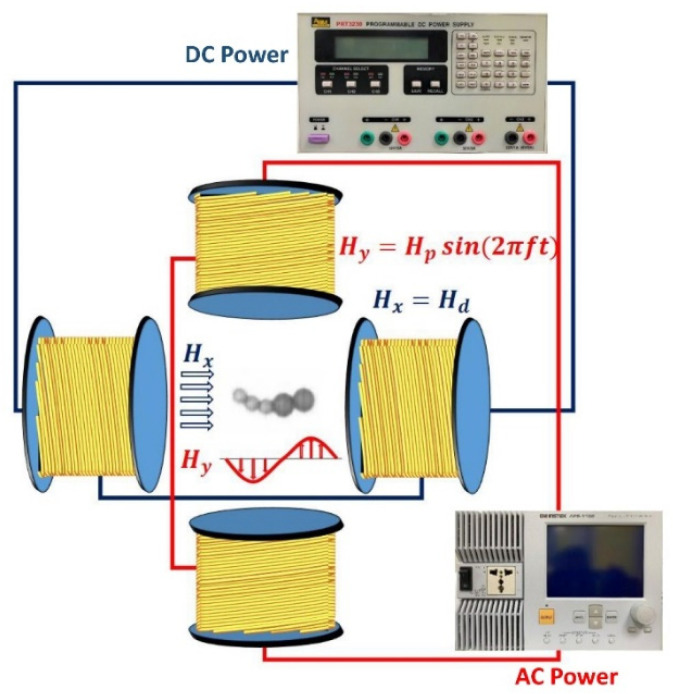
A schematic diagram of the experimental setup. A static directional magnetic field powered by a DC source was initially applied to create a linear microswimmer. An additional sinusoidal dynamic field from an AC power supply was then applied perpendicularly to generate an oscillating field, which created a waggling motion for the magnetic microswimmer.

**Figure 2 micromachines-11-01107-f002:**
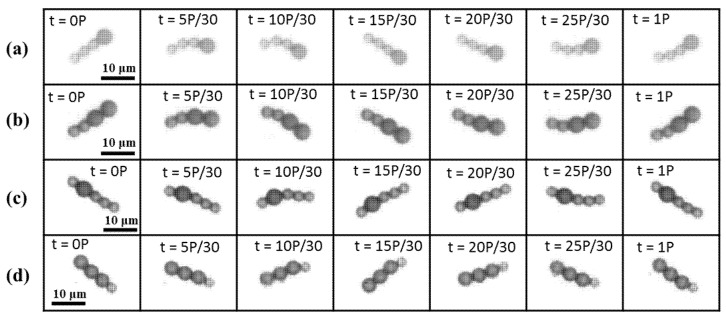
The evolution of the distortion of the (**a**) L1S3, (**b**) L2S2, (**c**) S1L1S3, (**d**) L3S1 swimmers in an oscillating field (*H*_p_ = 2000 A/m, *H*_d_ = 1900 A/m, *f* = 10 Hz). The images were captured within an arbitrary period (denoted as P) of oscillation. All swimmers oscillated clockwise from the top position and anticlockwise when they reached the bottom, and then returned to the initial point.

**Figure 3 micromachines-11-01107-f003:**
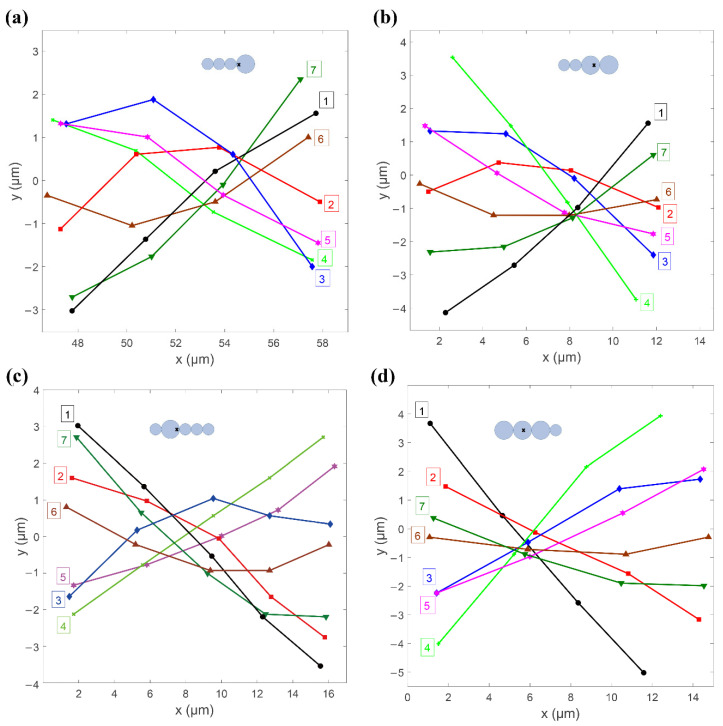
Waveform evolutions for the (**a**) L1S3, (**b**) L2S2, (**c**) S1L1S3, and (**d**) L3S1 swimmers shown in Figure 2. The dots mark the theoretical position of the center of mass of the swimmer, while the numbers represent the sequential waveforms of the swimmer within a full oscillating period. The interval of each waveform is 5P/30, as shown in the sequential images in Figure 2.

**Figure 4 micromachines-11-01107-f004:**
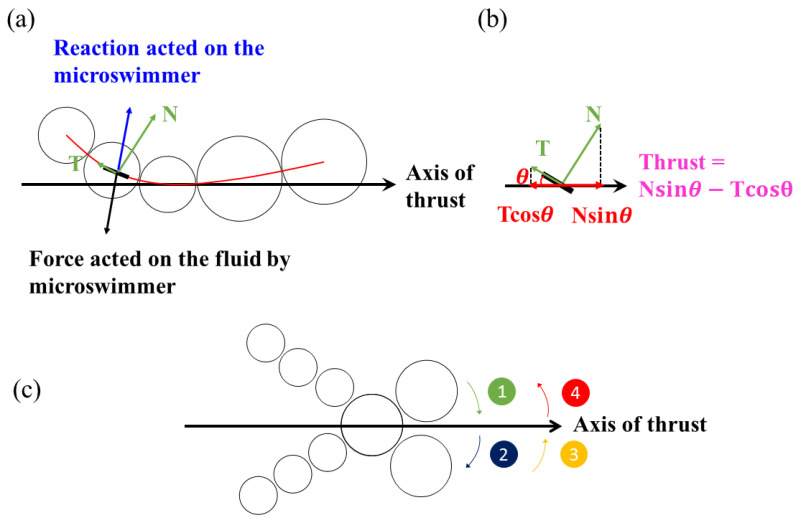
Diagram of the net thrust generation mechanism for a microbead swimmer in a low Reynolds number environment: (**a**) The forces that are imposed on an oscillating swimmer and (**b**) a schematic diagram of the net thrust resulting from the projection of the normal (N) and tangential (T) components along the axis of the thrust. (**c**) Schematic diagram of the four stages within a period of oscillation for a swimmer.

**Figure 5 micromachines-11-01107-f005:**
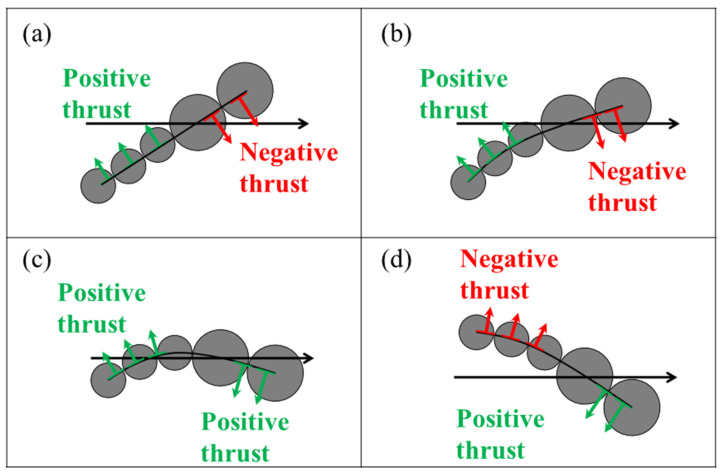
Schematic diagram of sequential images showing the direction of the thrust for the head and tail of a swimmer within the first half-period of oscillation. (**a**) Initial position. (**b**) The head and tail generate negative and positive thrust, respectively. (**c**) The head and tail simultaneously generate positive thrust. (**d**) The head and tail generate positive and negative thrust, respectively.

**Figure 6 micromachines-11-01107-f006:**
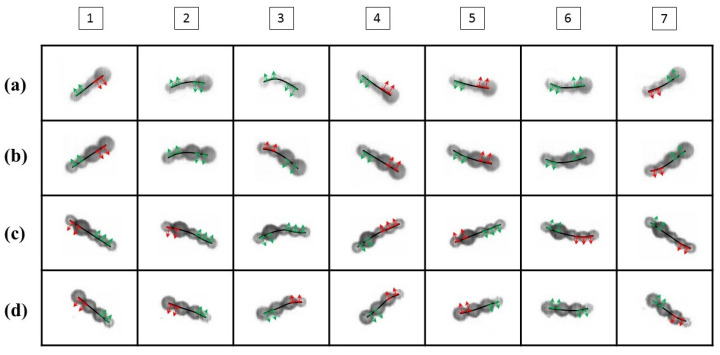
Sequential images of the direction of thrust for the head and tail of a swimmer within a period of oscillation for the (**a**) L1S3, (**b**) L2S2, (**c**) S1L1S3, and (**d**) L3S1 swimmers in Figure 2. The numbers at the top of the figure correspond to the number of the curve in Figure 3. The green and red arrows represent the positive and negative thrust, respectively, of the head and tail of the swimmer.

**Figure 7 micromachines-11-01107-f007:**
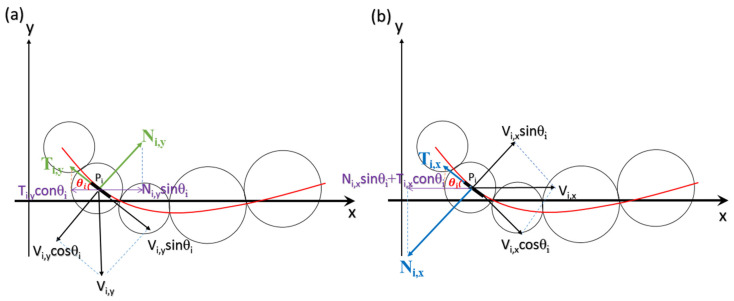
Schematic diagram of the forces imposed on the *i*th particle (**a**) when moving vertically along the axis of propulsion (x-axis) at the velocity of *V_i,y_*, or (**b**) when moving along the x-axis at the velocity of *V_i_,_x_*.

**Figure 8 micromachines-11-01107-f008:**
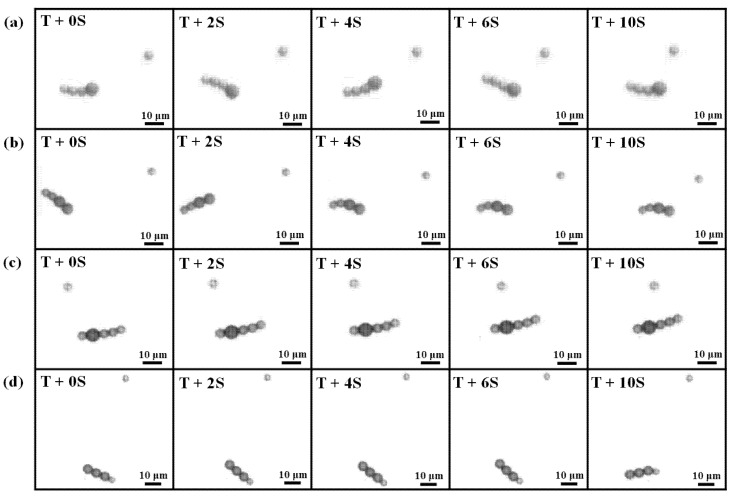
Sequential images of the (**a**) L1S3, (**b**) L2S2, (**c**) S1L1S3, (**d**) L3S1 swimmers in an oscillating field (*H*_p_ = 2000 A/m, *H*_d_ = 1900, A/m, *f* = 10 Hz). T represents the time when the swimmer began swimming at the highest speed and then cruised at a constant speed, and S represents second. A static single particle was used as a reference point to observe the locomotion of the swimmer, as well as for evidence of the uniformity of the applied field.

**Figure 9 micromachines-11-01107-f009:**
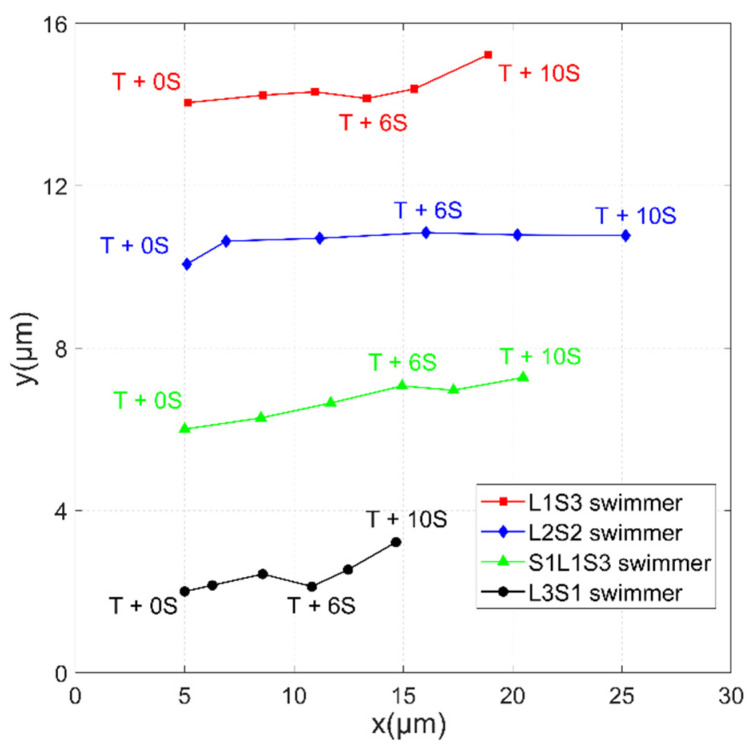
Trajectories of the swimmers in Figure 8 moving more 10 s.

**Figure 10 micromachines-11-01107-f010:**
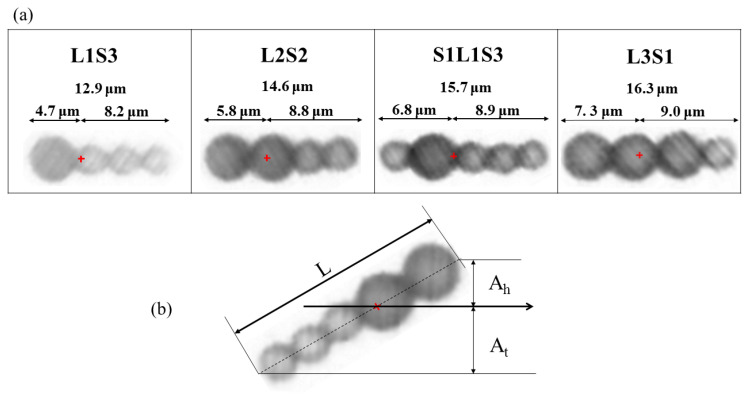
(**a**) Schematic diagram of the configuration of the different types of swimmers. (**b**) Definition of the oscillating amplitude of the head (*A*_h_) and the tail (*A*_t_) for a swimmer. *A*_h_ is the distance measured from the tip of the head particle to the center of the mass of the swimmer; *A*_t_ is the distance between the tip of the tail particle and the center of the mass of the swimmer; L represents the overall length of the swimmer.

**Figure 11 micromachines-11-01107-f011:**
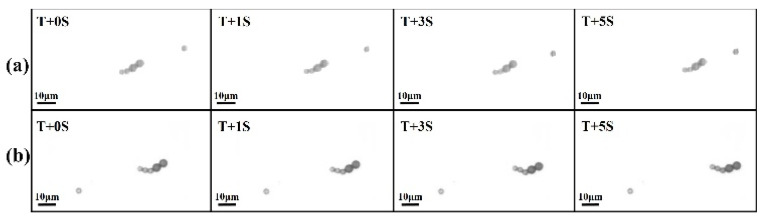
Sequential images of the (**a**) L2S2 and (**b**) L2S3 swimmers in an oscillating field with a strength of *H*_p_ = 1500 A/m and *H*_d_ = 1900 A/m and a frequency of *f* = 10 Hz. *T* represents the time when the swimmer started swimming at the greatest velocity and then cruised at a nearly constant speed, while *S* refers to second.

**Figure 12 micromachines-11-01107-f012:**
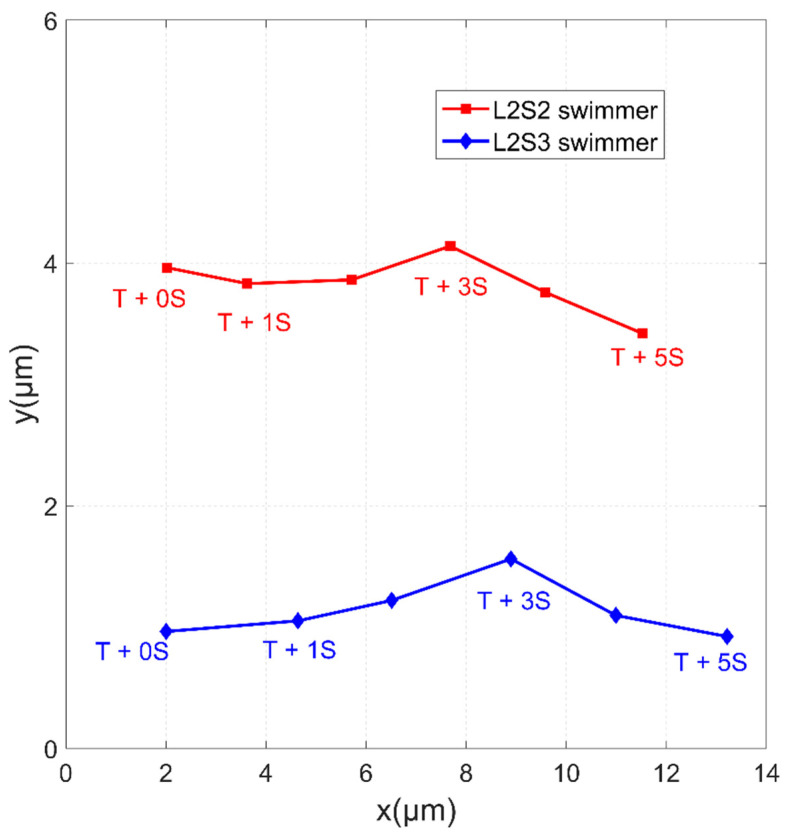
Trajectories for the swimmers shown in Figure 11 in a period of 5 s.

**Figure 13 micromachines-11-01107-f013:**
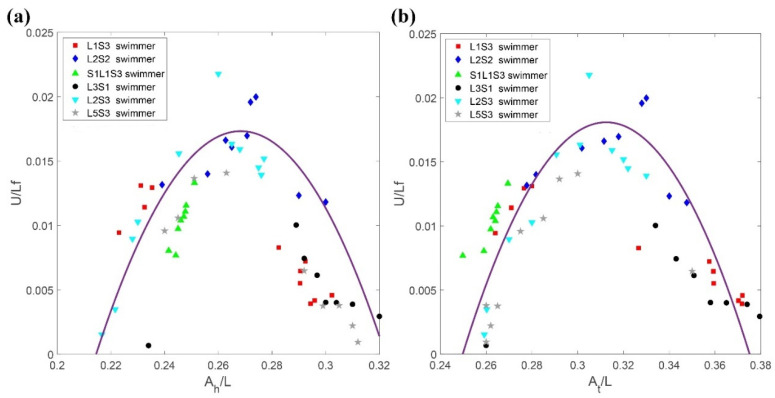
Distribution of the dimensionless swimming velocity in terms of the ratio of maximum amplitude of oscillation for (**a**) the head (*A*_h_) and (**b**) the tail (*A*_t_) to the length of the swimmers.

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
