# Peer review of "Propulsion Mechanism of Flexible Microbead Swimmers in the Low Reynolds Number Regime"

_micromachines, 2020, doi:10.3390/mi11121107_

Round 1
Reviewer 1 Report
This study is well-written and its content is very interesting. Therefore, I recommend the publication of this paper after the incorporation of the following comments:
Major comments:
[1] Fabrication and characterization: How do you ensure that the self-assembly of the microparticles chains give the desired configuration? What is the response of the chain when the applied magnetic field is removed? Is the assembly process reversible ?
[2] Magnetic actuation and frequency response: Provide frequency response results to show the influence of the actuation frequency on the swimming speed ? What is the step-out frequency of the swimmers ? What is the influence of the different configuration on the step-out frequency and on the range of the frequency response ?
[3] Theoretical framework: The authors provided nice schematics for the propulsion principle. However, these schematics do not lead the reader to calculate the propulsive thrust of any of the swimmers. For example, you need to show a formula for the net thrust force based on the presented framework. I understand that the experimental nature of the work. However, to complete the theory and to better present the propulsion mechanism, the authors are suggested to elaborate on the time-averaged net propulsive force for each group.
[+] State-of-the-art in soft microrobotics: Here the authors need to refresh the introduction with related work on the magnetic actuation of soft robotic sperms, biohybrid microrobots like the ironsperm, spermbots, and magnetosperm. These studies have been investigated in different settings and include relevant analysis. The state of the art can also be improved if the authors also include details about the resistive force theory which can be used to describe the propulsive force for a wide range of design variables.
[+] Magnetic setup and magnetic actuation: The proposed setup consists of four electromagnetic coils surrounding the workspace. Does the configuration generate magnetic field gradient ? If so, how do you make sure that the pulling force on the chains does not (or does) contribute to the motion of the chain ? I mean why you did not consider using a configuration that produce uniform magnetic fields only.
Reviewer 2 Report
The authors characterized the propulsion mechanism of a flexible magnetic microswimmer under an oscillating magnetic field. They obtained the relationship between dimensionless swimming velocities and the ratio of the oscillation amplitude and length of swimmers in Fig. 14. This graph seems valuable.
1) To manipulate magnetic particles, undulation is not necessary and an external magnetic field is enough. Why is this topic important?
2) What are the major differences from the following previous studies? It seems that the authors use the same experimental setup and study similar kinds of motions.
Biomicrofluidics Vol. 10, 011902 (2016); https://doi.org/10.1063/1.4939945
IEEE Transactions on Magnetics, Vol. 54, Issue: 11, (2018) DOI: 10.1109/TMAG.2018.2845915
3) The title of this paper sounds too general.
4) Variables and equations are not properly expressed in the whole text and it reduces the readability of this paper.
5) Extensive modifications of the English language and styles are also required even before submission.
6) Some figures are difficult to understand and need to be modified.
- Fig. 2 What's the meaning of P?
- Fig. 3 What do the numbers show? What's the interval of each point?
- Fig. 4-6 Why do the authors use five swimmers to explain the model of four and five swimmers?
- Fig. 7 Which type of swimmers is most similar to the model in Fig. 6?
- Fig. 8, 11 Would you tell the meaning of S?
- Fig. 9,12 How did you define the starting points of the swimmers?
- Fig. 13 How are the amplitudes of the swimmers are defined from this figure?
7) Why did the authors choose 10 Hz for the frequency?
8) Why were limited combinations of particles characterized?
Round 2
Reviewer 1 Report
The authors have incorporated all my previous comment. Thanks.
Minor comments:
1) if possible, please replace the schematics in Figure 11 with real microscopic images of the chains.
2) Combine Figure 14 and Figure 11. I think same information is repeated.
3) Combine Figures 4 and 5.
4) In the introduction, I think the author need to refer to reference [29] by the name(s) of the first author. Please check this part.
Reviewer 2 Report
The authors gave substantial responses to the reviewer. The reviewer's comments 4) and 5) have not been sufficiently addressed. Formatting variables and equations and English proofreading are necessary before publication. The other comments are well addressed.
